https://doi.org/10.1038/s41467-022-29299-0　　**OPEN**

# The promotion effect of π-π interactions in Pd NPs catalysed selective hydrogenation

Miao Guo[1,5], Sanjeevi Jayakumar[1,5], Mengfei Luo [2✉], Xiangtao Kong[3], Chunzhi Li[1,4], He Li [1], Jian Chen[2] & Qihua Yang [1,2✉]

The utilization of weak interactions to improve the catalytic performance of supported metal catalysts is an important strategy for catalysts design, but still remains a big challenge. In this work, the weak interactions nearby the Pd nanoparticles (NPs) are finely tuned by using a series of imine-linked covalent organic frameworks (COFs) with different conjugation skeletons. The Pd NPs embedded in pyrene-COF are ca. 3 to 10-fold more active than those in COFs without pyrene in the hydrogenation of aromatic ketones/aldehydes, quinolines and nitrobenzene, though Pd have similar size and surface structure. With acetophenone (AP) hydrogenation as a model reaction, systematic studies imply that the π-π interaction of AP and pyrene rings in the vicinity of Pd NPs could significantly reduce the activation barrier in the rate-determining step. This work highlights the important role of non-covalent interactions beyond the active sites in modulating the catalytic performance of supported metal NPs.

[1] State Key Laboratory of Catalysis, Dalian Institute of Chemical Physics, Chinese Academy of Sciences, Dalian 116023, China. [2] Key Laboratory of the Ministry of Education for Advanced Catalysis Materials, Zhejiang Key Laboratory for Reactive Chemistry on Solid Surfaces, Institute of Physical Chemistry, Zhejiang Normal University, Jinhua 321004, China. [3] College of Chemistry and Chemical Engineering, Anyang Normal University, Anyang 455000, China. [4] University of Chinese Academy of Sciences, Beijing 100039, China. [5]These authors contributed equally: Miao Guo, Sanjeevi Jayakumar. ✉email: mengfeiluo@zjnu.cn; yangqh@dicp.ac.cn

Inspired by biocatalysis, engineering the microenvironment of supported metal nanoparticles (NPs) to enhance their catalytic activity and selectivity has attracted a great amount of research interests[1–3]. The specific physicochemical microenvironment surrounding the metal NPs may significantly regulate their catalytic behaviour and is considered to be as important as the size, shape and metal-support interactions[4–6]. Hence, the fundamental understanding and the rational modulation of microenvironment nearby the metal NPs have become an important topic in the field of heterogeneous catalysis.

The assembly of organic ligands or polymers on metal surface is a general approach to construction a specific microenvironments in the vicinity of metal NPs[7–10]. The steric confinement effects created by surface modifiers can effectively improve the selectivity via modulating the adsorption mode of the substrates[11,12], regulating the coordination of reactants on catalytic centres[13] and even manipulating the binding strength of intermediates[14,15]. For instance, the selectivity of C=O hydrogenation in cinnamaldehyde and furfural hydrogenation can be greatly increased via modulating the adsorption mode of substrates on metal surface by construction of crowded "fence" on the surface of Pt or Pd NPs with densely self-assembled monolayers (SAM) modifiers[11,12]. The organic modifiers can also interact with reactants or intermediates to be hydrogenated and thus affect the overall catalytic performance. For example, the surface modified chiral ligands can promote the enantioselectivity in metal NPs catalysed enantioselective hydrogenations via the weak interactions (e. g. H-bond, π–π non-covalent interactions)[16,17].

The nanospace of porous materials is another desirable platform to modulate the weak interactions surrounding the metal NPs through the specific functional groups integrated in the porous materials[18–20]. The functional groups of porous material could regulate the interaction of the substrates[18,19] or the reaction intermediates[20], thereby affecting the activity and selectivity of the confined metal NPs. Recently, our group reported that the phosphine ligands in the vicinity of Ru NPs which preferentially interact with COOH of benzoic acid can behave like a "regulator", making inactive Ru NPs active in benzoic acid hydrogenation under mild conditions[19]. In most cases, the effect of microenvironment entangles with the alteration of the electronic and steric structure of metal NPs. Studies on using solely the weak interaction to modulate the catalytic performance of metal NPs have rarely been reported.

The principle challenging issue for utilization of weak interactions to regulate the catalytic performance of metal NPs is the lack of host materials with precisely designed organic moieties and pore structure at molecular level[21]. Imine-linked covalent organic frameworks (COFs), featuring abundant conjugated aromatics, imine bonds and functional side chains, could be an ideal host material to engineer the weak interactions surrounding the metal NPs[22–25]. Although the promotion effect of COFs has been recently reported in organic transformations[26] or for immobilized metal complex in asymmetric catalysis[27], the comprehensive understanding of the substrate-COFs weak interactions and their effects on the catalytic performance of metal NPs are still lacking.

In this work, the imine-linked COFs with different chemical compositions and topologic structures were employed to modulate the weak interactions surrounding Pd NPs. With almost identical surface geometric and electronic structures, Pd NPs hosted in pyrene-containing COF (Py-COF) are more active than those in COFs without the pyrene moiety in the selective hydrogenation of polar functional groups (C=O, C=N, NO₂, C=C–O) of aromatic compounds. The systematic thermodynamic, kinetic studies and DFT calculations imply that the π–π interactions of the substrates and pyrene rings orderly arranged in pyrene-COF can direct the preferential adsorption of carbonyl group on Pd surface and reduce the activation barrier in the rate-determining step (RDS) to enhance the activity.

## Results

**Synthesis and characterizations of Pd NPs confined in imine-linked COFs.** A new and stable pyrene-containing COF (Py-COF) was synthesized by the condensation of 1, 3, 6, 8-tetrakis (4-aminophenyl) pyrene (Py) and 2, 5-dimethoxyterephthalaldehyde (DMTA) (Fig. 1a). The characteristic C=N vibration at 1612 cm⁻¹ in the Fourier transform infrared spectroscopy (FT-IR) spectrum (Supplementary Fig. 1a) and chemical shifts at 157 ppm assigned to C=N in the solid-state ¹³C cross-polarization/total sideband suppression (¹³C-CP/TOSS) spectrum (Fig. 1b) indicate the successful formation of Py-COF with the imine linkage. The diffraction peaks at 3.8°, 5.8°, 7.6° and 23.4° in the PXRD pattern of Py-COF (Fig. 1c) can be assigned to the (110), (210), (300) and (001) facets, respectively. The simulated powder X-ray diffraction (PXRD) patterns of the eclipsed AA stacking conducted by Materials Studio Software are in good agreement with the experimental PXRD patterns with a C2/m space group (Supplementary Figs. 1c, d). The Pawley refinement cell parameters are $a = 36.927$ Å, $b = 31.363$ Å, $c = 4.047$ Å, $\alpha = 92°$, $\beta = 110°$ and $\gamma = 89°$, with good agreement factors $R_{wp} = 6.96\%$ and $R_p = 8.96\%$ for Py-COF. Py-COF has a microporous structure with a Brunauer–Emmett–Teller (BET) surface area of 1183 m² g⁻¹ and pore volume of 1.3 cm³ g⁻¹ (Fig. 1d). The pore size of Py-COF is mainly distributed at about 1.6 nm, which agrees well with the theoretical data (1.7 nm) calculated by the AA stacking model. The above result indicates that Py-COF has a well-stacked structure. Furthermore, almost identical PXRD patterns to the fresh sample were obtained after treatment of Py-COF in hot water, concentrated HCl and 14 M NaOH (Fig. 1e), suggesting the high chemical and structural stability of Py-COF. Py-COF can be stable up to 380 °C under air as revealed by thermogravimetric analysis (TGA), showing good thermal stability of Py-COF (Supplementary Fig. 1b).

Other imine-linked COFs (Be-COF, TB-COF) with different topological structures and compositions were synthesized according to the literature methods (Fig. 1a, Supplementary Fig. 2)[28,29]. The formation of imine-linked COFs was confirmed by FT-IR characterization (Supplementary Fig. 3a). The PXRD patterns show that Be-COF and TB-COF have high crystallinity with eclipsed AA stacking mode (Supplementary Fig. 3b), consistent with the reported results[28,29]. The Be-COF has BET surface area of 2076 m² g⁻¹, pore volume of 1.3 cm³ g⁻¹ and pore size of 2.8 nm and TB-COF has BET surface area of 1234 m² g⁻¹, pore volume of 0.73 cm³ g⁻¹ and pore size of 2.8 nm (Supplementary Figs. 3c, d). Be-COF and TB-COF both have higher BET surface area and larger pore sizes than those of Py-COF.

Py-COF, Be-COF and TB-COF with different conjugated skeletons were chosen as models to investigate the influence of non-covalent interactions on the catalytic performance of Pd NPs considering their different tendency to form π–π interaction with aromatic compounds[30]. The three COFs with the imine-linkage could be ideal supports for hosting ultrafine metal NPs due to the strong coordinating ability of imine with metal ions[31]. Pd NPs were loaded in COFs by the traditional wet impregnation-reduction method. High-resolution transmission electron microscopy (HRTEM) images show that Pd NPs embedded in the three COFs have a particle size of ~1.7 nm (Figs. 2a, c, Supplementary Fig. 4a). The high-angle annular dark-field scanning transmission electron microscopy (HAADF-STEM) images confirm the HRTEM results (Figs. 2b, d, Supplementary Fig. 4b). The

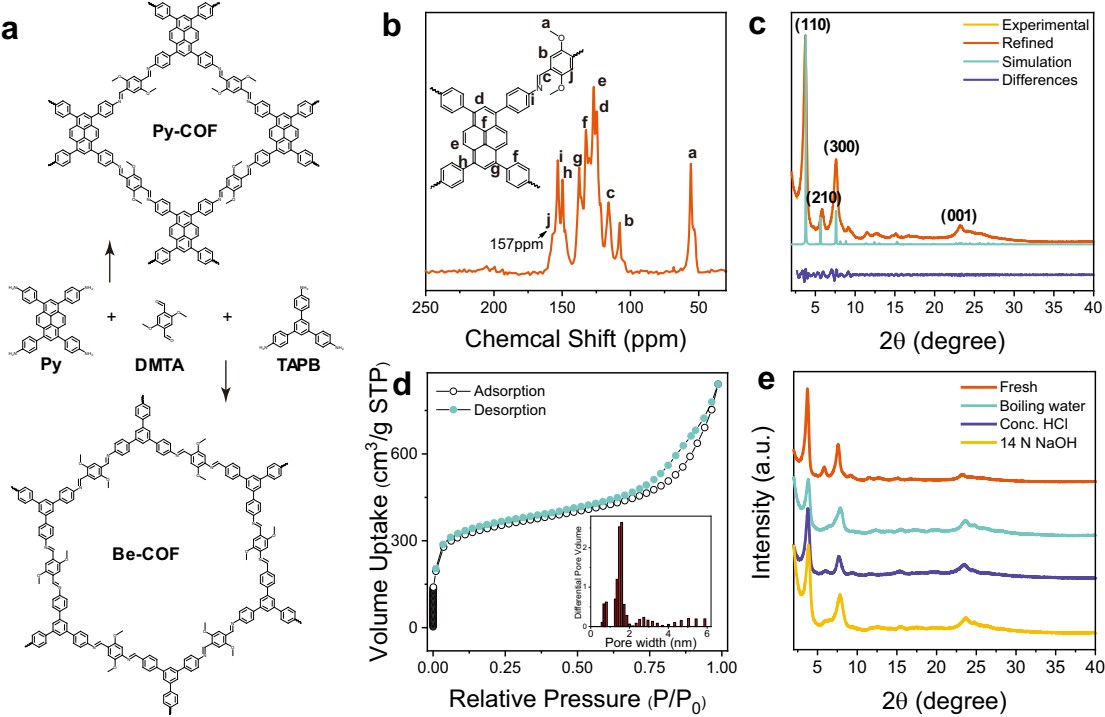

**Fig. 1 Synthesis, characterization and stability test of COFs. a** Synthesis of Py-COF and Be-COF by the condensation of DMTA with Py and TAPB, respectively. **b** $^{13}$C-CP/TOSS NMR spectrum, **c** Experimental, Pawley-refined and simulated PXRD patterns (AA stacking) and difference plot, **d** N$_2$ adsorption isotherm and pore size distribution and **e** PXRD patterns of Py-COF before and after treatment in boiling water, 12 N HCl and 14 N NaOH for 3 days.

energy-dispersive X-ray spectroscopy (EDS) results of the representative Pd/Py-COF show that the Pd NPs are in close contact with N and C, implying the existence of organic moieties in the vicinity of Pd NPs (Fig. 2e). It should be noted that the ultrafine Pd NPs of 1.7 nm on Py-COF could still be obtained with 10 wt% Pd (Supplementary Figs. 4c, d), showing the unique characters of COFs in stabilizing ultrafine metal NPs even at high metal loading amounts. The XRD patterns of Pd/COFs are similar with those of the parent COFs and no diffraction peaks assigned to Pd NPs could be observed, implying the well retained crystalline structure after Pd loading and the absence of large Pd particles (Supplementary Fig. 5). The Pd dispersion measured by CO adsorption is respectively 26.2%, 26.1% and 24.2% for Pd/Py-COF, Pd/Be-COF and Pd/TB-COF (Supplementary Table 1), showing that all the Pd/COFs have similar Pd surface area. The Pd dispersion obtained by H$_2$ chemisorption gave a similar trend with CO chemisorption. As compared, the dispersion derived from H$_2$ chemisorption was a little lower than that from CO chemisorption. This may be due to the diversified chemisorbed form or species of different adsorbates on Pd surface[32,33].

In situ FT-IR spectra of CO adsorption was first conducted to characterize the surface properties of Pd NPs confined in the COFs. As shown in Fig. 2f and Supplementary 6a, Pd/Py-COF, Pd/Be-COF and Pd/TB-COF show the linear-bonded CO at ~2045 cm$^{-1}$ together with the bridge-bonded CO at ~1915 cm$^{-1}$ (ref. [12]). It should be mentioned that the linear to bridge ratio (L/B ratio) for the three catalysts is almost the same (Supplementary Table 1), suggesting that Pd NPs in the three COFs may have similar electronic and surface geometric structures. The different peak intensity of Pd/Py-COF and Pd/Be-COF may be due to the different transmittance on the two samples with different colours (brown and yellow respectively for Pd/Py-COF and Pd/Be-COF). The identical Pd 3d binding energies (BEs) and Pd$^0$/Pd$^{2+}$ ratio obtained by X-ray photoelectron spectroscopy (XPS) characterization further confirm

that Pd NPs in the three COFs have similar surface electronic structure and reduction degree (Fig. 2g, Supplementary Fig. 6b, Supplementary Table 1). In comparison with Pd/C, the Pd 3d BEs of Pd/COFs show a red shift of 0.1 eV, showing the electron donation from imine to Pd NPs. As compared with conventional oxide supports which could transfer electrons to metal NPs, smaller shift in Pd 3d BEs may be related to bond character of N ligands which donates instead of transfer electrons to Pd NPs[34].

The electronic structure of Pd NPs was further investigated using the CO stripping voltammetry technique (Fig. 2h). The CO stripping potential of Pd/Py-COF and Pd/Be-COF respectively at 0.84 V and 0.85 V show the similar electronic structure of the two catalysts. The CO stripping potential of Pd/COFs is higher than that of Pd/C (0.72 V), suggesting the C–O bond weakening by the back-donation of electrons from electronic rich Pd/COFs to the 2π* antibonding orbitals of CO[35]. The similar electronic and geometric structure for the three catalysts is possibly related to 2D layered stacking structures of the COFs[36] and strong coordination of N with Pd than other functional groups in COFs[37].

**Promotion effect of π−π interactions in Pd catalysed hydrogenation.** On the basis of the above characterizations, Pd NPs in the three COFs with similar size, surface electronic/geometric structure but different chemical environments were used as model catalysts to investigate the effect of the non-covalent interactions on the catalytic performance of metal NPs. Considering the tilting adsorption configuration of carbonyl on the Pd surface[38] may induce the interaction of aromatic carbonyls with the conjugation skeleton of COFs, acetophenone (AP) hydrogenation was selected as the model reaction. AP hydrogenation with Pd NPs could produce phenyl ethanol and ethylbenzene under mild conditions (Fig. 3a). At 40 °C and 10 bar H$_2$, Pd/Py-COF could efficiently catalyse the hydrogenation of AP with >99% conversion and >99% selectivity to phenyl ethanol in

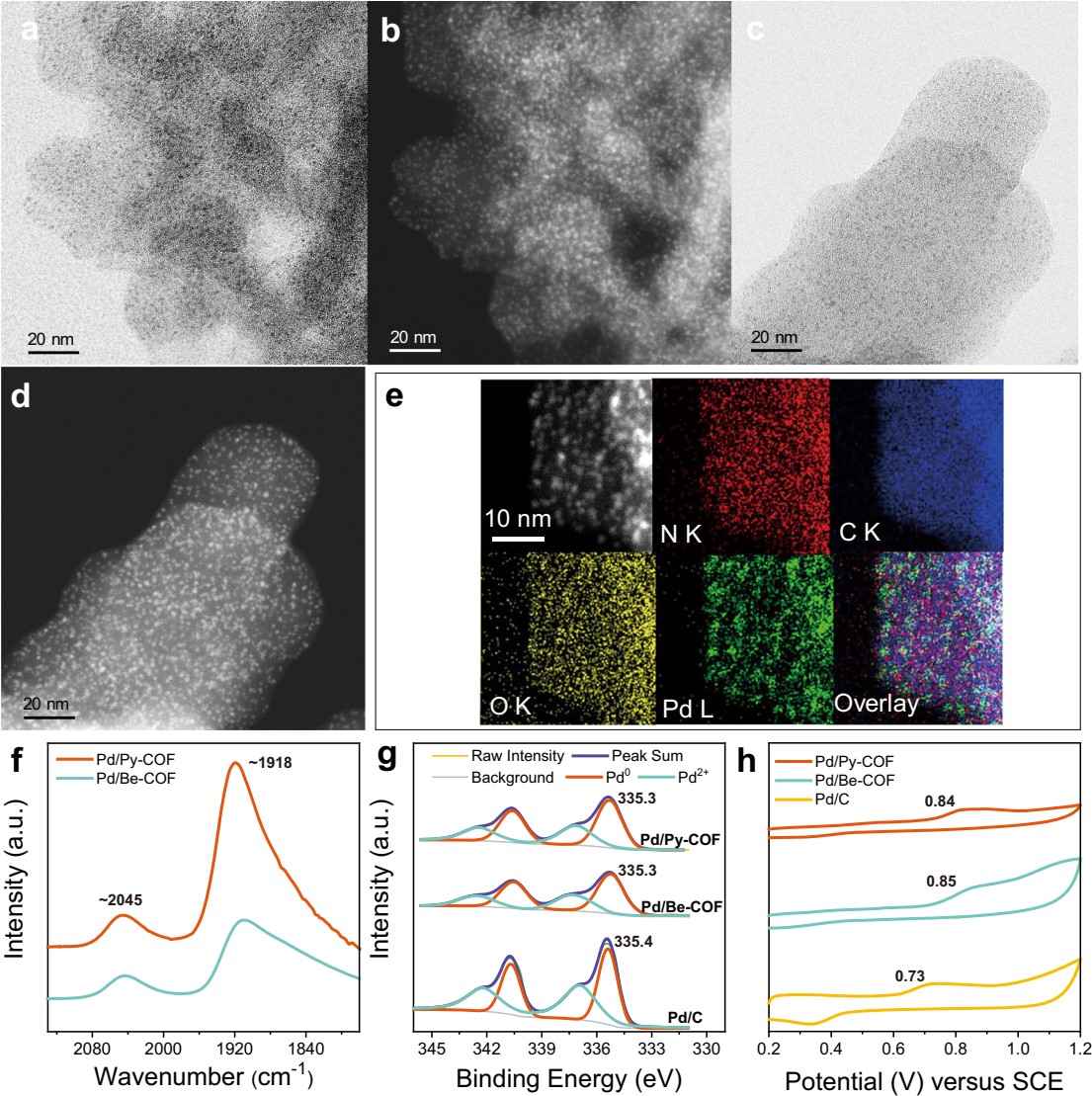

**Fig. 2 Characterization of supported Pd NPs. a** HRTEM image and **b** HAADF-STEM image of Pd/Py-COF. **c** HRTEM image and **d** HAADF-STEM image of Pd/Be-COF. **e** N, C, O, and Pd STEM-EDS maps and reconstructed overlay image for Pd/Py-COF. **f** In situ FTIR spectra of CO adsorption, **g** Pd 3d XPS core-level spectra and **h** CO stripping voltammetry for Pd/COFs and Pd/C.

3 h. Further prolonging the reaction time after 100% AP conversion, no conversion of phenyl ethanol to side products could be observed with Pd/Py-COF (Figs. 3b, c). Under identical conditions, Pd/Be-COF and Pd/TB-COF also exhibited high selectivity to phenyl ethanol (>99%) although the conversion was much lower than that of Pd/Py-COF (Fig. 3b, Supplementary Table 2). The apparent turnover frequency (TOF) calculated with ~10% AP conversion of Pd/Py-COF is almost 5.5 and 6.5-fold that of Pd/Be-COF and Pd/TB-COF, respectively. No direct relation of BET surface area, Pd particle size and surface structure with activity of Pd/COFs could be observed. In contrast to Pd/COFs, no phenyl ethanol was detected with Pd/C as the catalyst in 3 h and the main products were ethylbenzene and ethylcyclohexane (Supplementary Table 2), suggesting that the activity and selectivity of AP hydrogenation changed obviously via modulation of the microenvironment surrounding the Pd NPs.

To identify the key factors of microenvironment on catalytic performance of Pd NPs, H-D exchange experiment was first performed to characterize the $H_2$ dissociation ability of supported Pd NPs (Fig. 3d, Supplementary Table 3). Pd/Py-COF and Pd/Be-COF have almost the same HD formation rate, lower than that of

Pd/C. The high selectivity of Pd/COFs is partly related to the weakened Pd−H which suppresses the further hydrogenolysis of C–O bond of phenyl ethanol[38,39]. Considering the similar H-D formation rate of Pd/Py-COF and Pd/Be-COF, the different catalytic activity of the two samples is not related to the $H_2$ dissociation ability. To further elucidate the activity difference of Pd/Py-COF and Pd/Be-COF, 1-aminopyrene (Py-NH$_2$) and aniline (AN) were respectively added in AP hydrogenation using commercial Pd/C as model catalyst (Fig. 3e). This is one of the most frequently used methods to tune the microenvironment of metal NPs[7,10]. In the presence of 0.02 equiv. of Py-NH$_2$ with respect to the surface Pd atom, the conversion of AP increases sharply from 18.1% to 29.1%. The conversion reaches a maximum of 34.5% with 0.13 equiv. of Py-NH$_2$. The typical volcanic curve shows the positive effect of pyrene ring in enhancing the AP activity of Pd/C. The decrease in AP conversion at higher amounts of Py-NH$_2$ is due to the active sites capping. In contrast, the AP conversion decreases obviously in the presence of AN. The above results suggest that the π–π interaction could boost the AP activity of Pd/C, which further confirms that π–π interaction enhances the activity of Pd/Py-COF

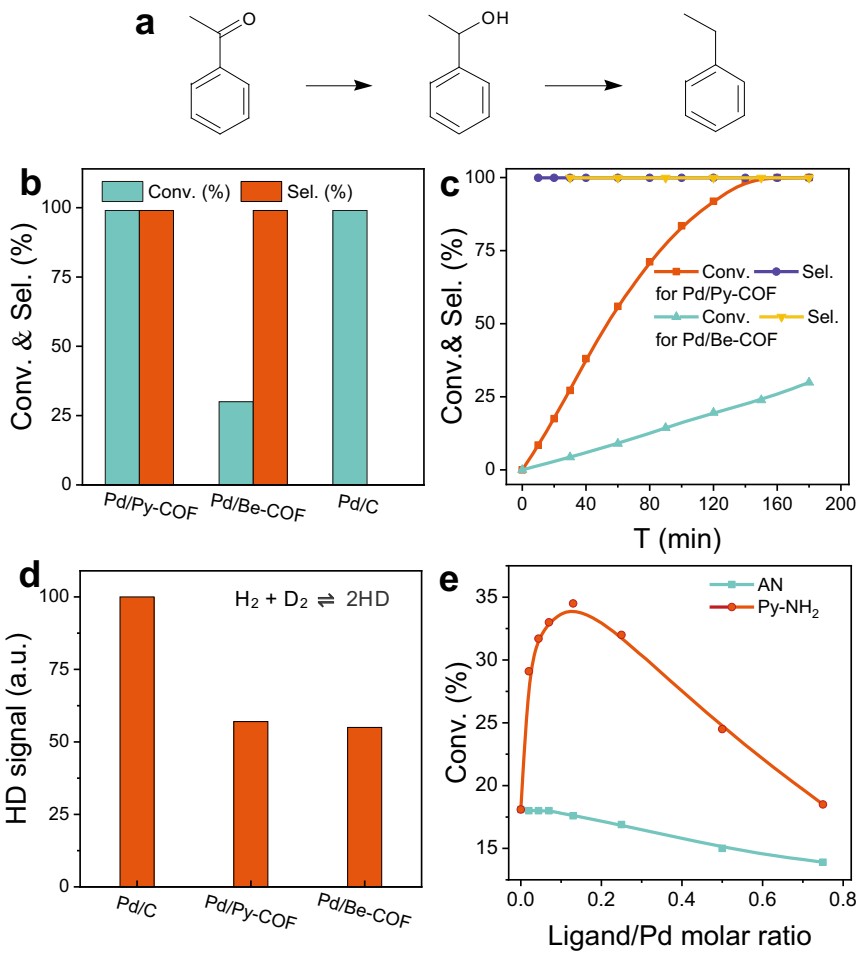

**Fig. 3 AP hydrogenation on supported Pd NPs. a** Reaction network of AP hydrogenation on Pd NPs. **b** The comparison of the conversion and phenyl ethanol selectivity and **c** reaction profiles of supported Pd NPs in AP hydrogenation. **d** H-D exchange results of supported Pd NPs. **e** Conversion on Pd/C with different amounts of 1-aminopyrene (Py-NH$_2$) and aniline (AN) in AP hydrogenation (Reaction conditions: 40 °C, 10 bar of H$_2$, 2 mL of EtOH, for **b**, **c**: 0.3 mmol of AP, Pd catalysts 0.833 mol%, for **e**: 1 mmol of AP, Pd catalysts 0.25 mol%).

in AP hydrogenation. A slightly increased selectivity to phenyl ethanol (~97% vs. 99%) for Pd/C after Py-NH$_2$ and AN modification was observed, similar to previous reports that the amine ligands favoured high selectivity to the alcohol in carbonyl hydrogenation[38,40].

With Pd/Py-COF and Pd/Be-COF as the model catalysts, the reaction orders toward H$_2$ and AP were measured. As shown in Fig. 4a, Pd/Py-COF and Pd/Be-COF have similar reaction orders with respect to H$_2$ (+0.14 and +0.13). According to the dual-site Langmuir–Hinshelwood mechanism, the approximately half and first order in H$_2$ at low H$_2$ pressure suggests the first hydrogenation and the second hydrogenation is the rate-determining step (RDS), respectively (Eq. 6–7 in the Supplementary Methods). Density functional theory (DFT) calculation demonstrates that the sequential hydrogenation of the O atom followed by C atom in carbonyl group has much lower free energy barrier than the opposite sequence (Fig. 4b, 0.198 eV vs. 0.631 eV), consistent with the kinetic results[41] and the recently reported DFT calculation[42]. Therefore, the RDS for Pd/Py-COF and Pd/Be-COF is the same and involves the O hydrogenation step. The apparent activation energies of AP hydrogenation were respectively 51.1 kJ/mol and 60.2 kJ/mol for Pd/Py-COF and Pd/Be-COF (Fig. 4c). The above thermodynamic and kinetic studies suggest that Pd/Py-COF and Pd/Be-COF with the same reaction pathways have different activation energy for AP hydrogenation.

DFT calculations were carried out to clarify the different activation barriers of Pd/Py-COF and Pd/Be-COF using a simple model with Pd$_4$ cluster loaded on Py-COF and Be-COF. It is very hard to accurately calculate the π–π interaction promoted AP hydrogenation on Pd surface due to the complex reaction system. To simplify the calculation, a Pd$_4$ was loaded on Py-COF and Be-COF, respectively. Because of the long distance of the aromatic group from the reactant in our Pd$_4$ models, the pyrene/benzene ring cannot directly influence the reaction process. However, the DFT calculation results show that the energy barrier of O addition step of AP hydrogenation could be significantly decreased via the interaction between phenyl ring and Pd atom. As shown in Supplementary Fig. 7, the dissociated hydrogen atoms were subsequently transferred to the oxygen atom of C=O group by overcoming barriers of 1.00 eV (transition state, TS) over Pd$_4$/Py-COF. However, a higher O hydrogenation barrier (1.35 eV) was observed for Pd/Be-COF. The result suggested that the TS stabilized by Pd via the phenyl ring of AP could significantly decrease the energy of O hydrogenation step. This also indirectly suggests that AP hydrogenation on Pd/Py-COF is more favourable than that on Pd/Be-COF, possibly due to stabilization of the TS via π-π interaction between pyrene rings and AP. The orders with respect to AP is respectively ~0 and +0.36 for Pd/Py-COF and Pd/Be-COF (Fig. 4a), indicating the stronger adsorption of AP on Pd/Py-COF than that on Pd/Be-COF[43]. We infer that the strong interaction results from the

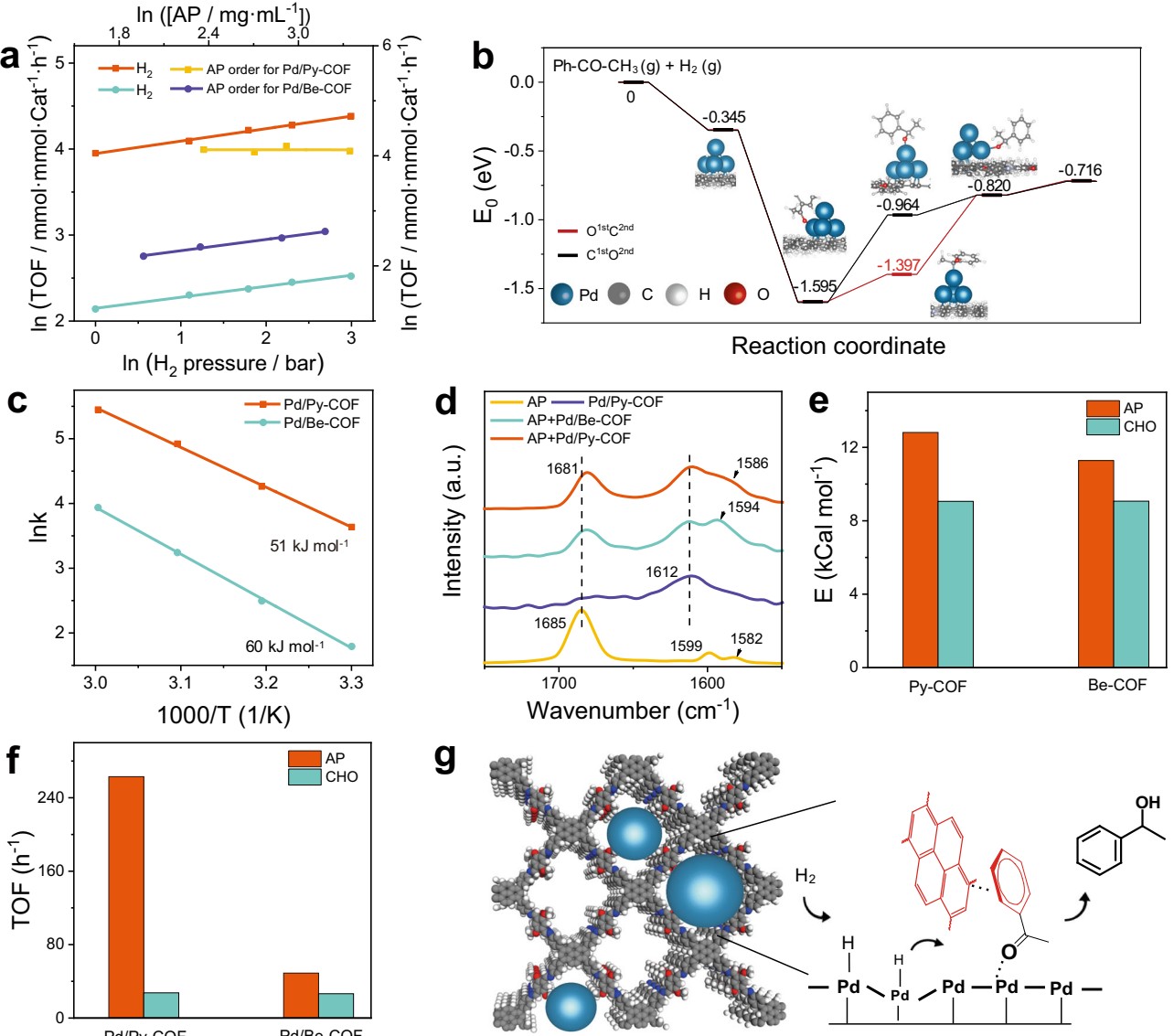

**Fig. 4 Mechanism investigation. a** Reaction orders with respect to H$_2$ and AP on Pd/Py-COF and Pd/Be-COF. **b** Potential energy profiles for carbonyl hydrogenation of AP with different sequences (O1st C2nd and C1st O2nd). **c** Arrhenius plots showing apparent activation barriers for Pd/Py-COF and Pd/Be-COF. **d** FT-IR spectra of AP, Pd/Py-COF and AP absorbed on Pd/Py-COF and Pd/Be-COF. **e** The interaction energies between COFs and AP/CHO. **f** TOFs of AP and CHO hydrogenation on Pd/Py-COF and Pd/Be-COF. **g** Schematic diagram of Pd NPs confined in Py-COF and proposed reaction mechanism of AP hydrogenation on Pd/Py-COF. (Reaction conditions for **a**: 40 °C, 2 mL of EtOH, H$_2$ pressure varies from 1 to 20 bar, AP concentration varies from ~7 to ~28 mg/mL. The TOF values for Arrhenius plots in **c** were recorded at 30, 40, 50 and 60 °C. Reaction conditions for **f**: 130 °C, 20 bar of H$_2$, 0.07 mmol of CHO, Pd catalysts 4 mmol%, 2 mL of H$_2$O).

direct preferential adsorption of carbonyl group on the Pd surface via the π–π interaction between pyrene rings and AP.

To further identify the interaction between AP and COFs, we performed the IR spectra for adsorption tests of AP on the Pd/Py-COF and Pd/Be-COF catalysts (Fig. 4d, Supplementary Fig. 8). The new IR band of C=O vibration at 1681 cm$^{-1}$ appears for Pd/Py-COF and Pd/Be-COF with adsorbed AP. In comparison with AP, the red shift of the C=O band from 1686 cm$^{-1}$ to 1681 cm$^{-1}$ indicates the coordination of C=O with surface Pd. Though the aromatic C-H vibrations of AP at 1599 cm$^{-1}$ and 1582 cm$^{-1}$ were covered by the aromatic C=C vibrations of Pd/Be-COF (Supplementary Fig. 8), the merged vibration peak for AP absorbed on Pd/Be-COF indicates that the activities of these vibrational modes are reduced/modified due to interaction of AP and Pd/Be-COF[44]. A broad band of aromatic C-H vibration of AP at ~1586 cm$^{-1}$ could also be clearly observed on Pd/Py-COF.

In comparison with AP on Pd/Be-COF, the red shift of aromatic C-H vibration of AP on Pd/Py-COF implies stronger interaction between AP and Pd/Py-COF[45]. The calculated interaction energies of AP with Py-COF are much higher than that with Be-COF (12.81 vs. 11.8 kcal mol$^{-1}$, Fig. 4e, Supplementary Figs. 9a, b), consistent with the previous reports that pyrene is easy to form π–π interactions with aromatics due to its planar conjugated electronic structure[30]. This is also confirmed by an obvious broadening of the aromatic H signals in $^1$H-NMR spectrum of AP in the presence of pyrene but not in the presence of mesitylene (Supplementary Fig. 10).

With an aim to further prove the accelerating effect of π–π interactions in AP hydrogenation, the activity of Pd/Py-COF and Pd/Be-COF was tested in the hydrogenation of cyclohexanone (CHO) and n-butylaldehyde which have similar carbonyl group to AP but without aromatic ring with H$_2$O as solvent (Fig. 4f).

Pd/Py-COF and Pd/Be-COF could efficiently catalyse the CHO and n-butyraldehyde hydrogenation to corresponding alcohols with >99% selectivity. Notably, Pd/Py-COF and Pd/Be-COF afforded almost the same TOFs (Supplementary Fig. 11, 28 vs. 27 h$^{-1}$), showing that the two catalysts have similar activity in CHO hydrogenation. The almost coincided kinetic curves of Pd/Py-COFs and Pd/Be-COF were observed for n-butyraldehyde hydrogenation, similar to CHO hydrogenation (Supplementary Fig. 11). The above results are quite different from the activity sequence in AP hydrogenation. The calculated interaction energies of CHO with Py-COF and Be-COF are similar (9.06 vs. 9.07 kcal mol$^{-1}$, Fig. 4e, Supplementary Figs. 9c, d), signifying the important role of π–π interactions in AP adsorption. Based on all the experimental and theoretical results, the relative strong π–π interactions of Py-COF and AP reduce the activation barrier in the RDS involving the O hydrogenation step, which promote both the activity and selectivity in AP hydrogenation to phenyl ethanol (Fig. 4g). This process was then boosted by the rapid desorption of phenyl ethanol considering that the adsorption of C=O on Pd surface is stronger than that of hydroxyl group[46].

**Generality of the activity promotion effect of π-π interactions.** The generality of the activity promotion effect of π–π interactions was tested in the hydrogenation of a series of aromatic carbonyls with Pd/Py-COF and Pd/Be-COF as model catalysts (Table 1,

Entries 1–3). In addition to AP, Pd/Py-COF could efficiently catalyse the hydrogenation of 2-acetonaphthone and 2-naphthaldehyde to the corresponding alcohols with >99% conversion and >99% selectivity. In comparison with AP, it needs longer reaction time to reach >99% conversion for 2-acetonaphthone, which is related with more rigid and conjugated structure of 2-acetonaphthone. The activity of Pd/Py-COF is more enhanced for the hydrogenation of 2-acetonaphthone than that for AP (6.1 vs. 5.5-fold as high as that of Pd/Be-COF), implying that stronger π-π interactions favour higher activity. As for 2-naphthaldehyde, Pd/Py-COF is 3.3-fold as active as Pd/Be-COF. Interestingly, we also found that Pd/Py-COF is more active than Pd/Be-COF in the hydrogenation of benzofuran to 2, 3-dihydrobenzofuran (5.6-fold), quinoline to 1, 2, 3, 4-tetrahydroquinoline (2.9-fold), N-benzylidenemethylamine to N-benzylmethylamine (2.9-fold) and nitrobenzene to aniline (4.0-fold, Table 1, Entries 4–7). For the hydrogenation of furfural to tetrahydrofurfuryl alcohol, the activity of Pd/Py-COF is more than one order that of Pd/Be-COF (439 vs. 38 h$^{-1}$, Table 1, Entry 8). Furthermore, we also tested the activity of Pd/Py-COF and Pd/Be-COF in the hydrogenation of styrene to ethylene benzene. Different from the above substrates, Pd/Py-COF with 8090 h$^{-1}$ TOF is less active than Pd/Be-COF with 12830 h$^{-1}$ TOF, indicating that the π–π interaction does not promote the hydrogenation activity of non-polar groups. This may be related with the different reaction mechanism and adsorption modes of

## Table 1 Comparison of the activity of Pd/Py-COF and Pd/Be-COF in the hydrogenation reactions[a].

| Entry | Substrate | Product | t (h) | Conv. (%) | Sel. (%) | TOF (h$^{-1}$)[b] | Ratio of TOF[c] |
|-------|-----------|---------|-------|-----------|----------|------------------|-----------------|
| 1 | | | 3 | >99 | >99 | 275 (50) | 5.5 |
| 2 | | | 10 | >99 | >99 | 92 (15) | 6.1 |
| 3 | | | 1 | >99 | >99 | 641 (195) | 3.3 |
| 4 | | | 3 | >99 | >99 | 277 (50) | 5.6 |
| 5 | | | 2.5 | >99 | >99 | 389 (134) | 2.9 |
| 6 | | | 0.75 | >99 | >99 | 15751 (5529) | 2.9 |
| 7 | | | 1 | 98 | 99 | 24015 (6015) | 4.0 |
| 8 | | | 1 | 52 | >99 | 439 (38) | 11.6 |

[a]Reaction conditions: 40 °C, 10 bar of H$_2$, 2 mL of EtOH, entries 1–5, 8: 0.3 mmol of substrates, Pd catalysts 0.833 mol%; entry 6: 8 mmol of substrates, Pd catalysts 0.05 mol%; entry 7: 8 mmol of substrates, Pd catalysts 0.02 mol%. Data in parentheses refer to Pd/Be-COF.
[b]TOF was calculated with a conversion <30%.
[c]The ratio of TOF of Pd/Py-COF and of Pd/Be-COF.

polar and non-polar groups on the metal surface[47]. The above results suggest that the π–π interactions could promote the activity of Pd NPs in the hydrogenation of polar functional group of aromatic compounds.

We also tested the recycling stability of Pd/Py-COF with AP hydrogenation as the model reaction (Supplementary Fig. 12). After the reaction, the catalyst recovered by centrifugation was washed thoroughly with ethanol and dried under vacuum. The recovered catalyst was directly used for the next run. Pd/Py-COF could be stably recycled for 4 cycles without obvious loss of catalytic activity and selectivity. After the fourth cycle, no aggregation of Pd NPs could be observed in the TEM image of used Pd/Py-COF (Supplementary Fig. 12b), demonstrating the high recycling stability of Pd/Py-COF.

## Discussion

Pd/Py-COF showed much higher activity than Pd/Be-COF and Pd/TB-COF in the hydrogenation of AP to phenyl ethanol, though the three catalysts have similar Pd size and surface structure. The reaction mechanism investigations and DFT calculations suggest that the activation barriers in the RDS involving the O hydrogenation step is reduced by the π–π interaction among pyrene and AP. In addition to AP hydrogenation, pyrene rings in the skeleton of COFs can greatly improve the activity of Pd NPs in the selective hydrogenation of a series of polar groups (C=O, NO$_2$, C=N) of aromatic compounds. In contrast, the activity promotion effect of pyrene cannot be observed in the hydrogenation of CHO to cyclohexanol, confirming the important role of the non-covalent π–π interactions in the activity promotion. The promotion effect of the π–π interactions in the catalytic activity signifies the importance of the weak interactions in modulation of the catalytic performance of heterogeneous catalysts beyond the active sites.

## Methods

**Preparation of Py-DMTA-COF (Py-COF)**. Specifically, 0.5 mmol of 1, 3, 6, 8-tetrakis (4-aminophenyl) pyrene (Py) and 1 mmol of 2, 5-dimethoxyterephthalaldehyde (DMTA) were dissolved in a mixture of solvents containing 10 mL of o-dichlorobenzene and 5 mL of 1-butanol. 1 mL 6 M CH$_3$COOH was successively added into the mixture (the solvent screening see Supplementary Table 4). After sonification for 2 min, the mixture was heated at 85 °C for 120 h under a N$_2$ atmosphere. The solid product was recovered by filtration and washed with copious amounts of methanol, followed by Soxhlet extraction using THF for 24 h. After drying under a vacuum at 60 °C for 6 h, the orange solid was obtained with a yield of 91%. The material was denoted as Py-COF. Elemental analysis calculated for Py-COF, C (81.6%), H (4.8%), N (6.4%). Found, C (79.6%), H (5.0%), N (6.5 %). The control materials, TPB-DMTA-COF (Be-COF) and BND-TFPB-COF (TB-COF) were prepared according to the literature reports[28,29].

**Preparation of Pd/Py-COF**. Typically, Pd/Py-COF with 5 wt% Pd loading was prepared by a traditional impregnation-reduction method. 95 mg of Py-COF was dispersed in 4 mL of deionized water, followed by the addition of 0.5 mL of Na$_2$PdCl$_4$ aqueous solution (0.01 g mL$^{-1}$). After stirring for 3 h, the solid was separated by filtration and washed with water. After drying at 100 °C for 30 min, the orange–red powder was reduced at 200 °C in H$_2$ for 2 h (2 °C min$^{-1}$). The preparation method of Pd/Be-COF and Pd/TB-COF was similar with Pd/Py-COF except that the Be-COF/TB-COF was used.

**Catalytic hydrogenations**. The hydrogenation reactions were carried out in a stainless-steel autoclave (300 mL) with a thermocouple-probed detector. Typically, a desired amount of the solid catalyst (2.5 × 10$^{-3}$ mmol Pd) was placed in an ampule tube, followed by the addition of AP (0.3 mmol) (S/C = 120) and 2 mL EtOH. The ampule tube was loaded into the reactor. After the tube was purged six times with hydrogen, the final pressure was adjusted to 10 bar and the reactor was heated to 40 °C with vigorous stirring. To ensure the collection of reliable catalytic data, the experiments were carried out at different stirring rates and the results showed that the reaction rate remained constant in the range of 800–1000 rpm, suggesting no influence of the external diffusion. The solvent was also optimized and the results show that EtOH is the best solvent for AP hydrogenation among the investigated protic and aprotic solvents (Supplementary Table 5). After the reaction, the solid catalyst was separated by centrifugation and the filtrate was collected, diluted with EtOH and analyzed by an Agilent 7890B GC equipped with an Agilent J&W GC HP-5 capillary column (30 m × 0.32 mm × 0.25 μm). Conversion and selectivity were determined using n-decane as the internal standard and the carbon balance of all the reactions is ~100%. The calculation method for conversion, selectivity and TOF was described in detail in the supplementary information. In the recycling test of AP hydrogenation, the catalyst recovered by centrifugation was washed with EtOH and used in the next run directly.

## Data availability
The data that support the findings of this study are available from the corresponding author under reasonable request. Source data are provided with this paper.

## Code availability
The authors did not use any previously unreported custom computer code or algorithm.

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

## Acknowledgements

We acknowledge financial support from the National Key R&D Program of China (2017YFB0702800), the National Natural Science Foundation of China (21733009), and the Strategic Priority Research Program of the Chinese Academy of Sciences (XDB17020200). We also thank Ms Peifang Yan from the Dalian Institute of Chemical Physics, CAS, for the experiment discussion.

## Author contributions

M.G. was responsible for most of the investigations, methodology development, data collection and analysis, and writing the original manuscript. S.J. designed and synthesized the Py-COF materials. X.K. carried out the part of the density functional theory calculations. C.L. and H.L. synthesized the Be-COF and TB-COF. J.C. and M.L. discussed the data. Q.Y. was responsible for the conceptualization, funding and revising of the manuscript.

## Competing interests

The authors declare no competing interests.
