## [Peer Review File · Nature Communications]

REVIEWER COMMENTS

Reviewer #1 (Remarks to the Author):

In this manuscript, Yang, Luo and coworkers described Pd/Py-COF showed much higher activity than Pd/Be-COF and Pd/TB-COF in hydrogenation of AP to phenyl ethanol for the π - π interaction among pyrene and AP. Pyrene rings in the skeleton of COFs can greatly improve the activity of Pd NPs in the selective hydrogenation of a series of polar groups (C=O, NO₂, C=N) of aromatic compounds. Moreover the characterizations of Pd NPs and a plausible mechanism were proposed. Hence, I would like to recommend this manuscript published in Nature Communications after the authors addressed the following issues:

- 1, As the authors mentioned that the effect of pyrene group in hydrogenation of cyclohexanone to cyclohexanol was found by calculation, so it indicated the important role of π - π interactions play in this system. Except the calculation, cyclohexanone or one more example should be tested under laboratory conditions to further confirm the effect.
- 2, Compare with entry 1 and 2 in Table 1: the strong interaction existed between pyrene rings and naphthalene ring, but it took longer time to access the final product? The similar phenomenon was found in entry 2 and 3. How about using the same reaction time to carry out these reactions?
- 3, For entry 7, the hydrogenation of was similar to be more difficult by using the Pd catalysts the authors made, but the author did large scale to evaluate the efficiency of Pd/Be-COF. Providing the results by decreasing the amount of substrate to 0.3 mmol? the reaction condition of entry was missing?
- 4, Many hydrogenation catalysts were known, I suggest to conduct a positive control to further confirm the advantages of Pd/Be-COF analogues.

Reviewer #2 (Remarks to the Author):

This paper demonstrates the importance of weak π - π interactions between the reactant and pyrene-containing COF (Py-COF: This also acts as a support of Pd nanoparticles) that would accelerate the catalytic selective hydrogenation performances on Pd nanoparticles. The authors insist this promotion effect of π - π interactions by presenting several control experiments using Be-COF and TB-COF that can possess Pd nanoparticles of similar size and surface structure. The catalytic reaction results are interesting; however, I have several important questions/comments that need to be addressed, which is related to the authors insist. Therefore, I cannot recommend acceptance in its current form in Nature Communications.

(1) line 204-208. The authors assume that one possible reason for the stabilization of TS of Pd/Py-COF on O hydrogenation step is the π - π interaction between pyrene ring and acetophenone (AP). However, it is very difficult to figure out the modelled structure of Pd/Py-COF and Pd/Py-COF to see the π - π interaction. The authors should present all calculated structures so as the reader can understand this interaction.

(2) line 213-225. The evidence of π - π interaction between the reactant and Py moiety of Py-COF seems to be weak; only FT-IR wavenumber shift of AP is discussed without appropriate literature citation and/or support of theoretical calculations. The authors calculated interaction energies between AP and Py-COF; however, there was also no appropriate literature citation.

(3) line 234-247. The comparison of catalytic reaction data using cyclohexanone (CHO) without aromatic ring is very important to experimentally support the effect of π - π interaction. TOFs of CHO on Pd/Py-COF and Pd/Be-COF were similar; however, the experimental conditions are the same to those on AP? The experimental conditions and TOF calculation method should be explicitly presented. How about conversion/selectivity plot similar to Figure 3b? The color bars of TOF value for CHO hydrogenation and interaction energy on Figure 4d are very similar, and the authors should revise it. Also how about other ketones with aliphatic hydrocarbon chains?

(4) line 191-194, Figure 4b. It is difficult to figure out the calculation routes (O hydrogenation followed by C, or opposite sequence). The authors must improve the Figure 4b by adding several key structural images.

(5) Characterization of Py-COF: The authors must present detailed characterization data if Py-COF is the new material. For instances, elemental analyses data in the experimental section.

(6) Overall: Figures (both in the main text and SI) are basically difficult to figure out. The authors have to make a great effort to improve the Figures so as to make readers easy to understand. For instances, peak attribution of FT-IR and peak legends XPS peak fitting are missing in Figure 2. Also, appropriate figure captions are required in some figures.

(7) HRTEM of only Pd/Py-COF is presented in Figure S4, which is easy to understand Pd nanoparticles. It is difficult to figure out the Pd nanoparticles of 1.7 nm when Pd loading is 10 wt% from Figure S4b. The authors should present HRTEM images for other samples.

(7) Experimental section: The equations to calculate conversion, selectivity, and TOF should be presented. The calibration method of the binding energy in XPS data must be presented. Experimental method for Table 1 should also be presented.

(8) Notation that is not an abbreviation should be presented at the first appearance. For instances TOSS appears at line 83 without no full notation. The authors must check thoroughly the manuscript.

Reviewer #3 (Remarks to the Author):

The results of studying Pd/COF hydrogenation catalysts containing different organic moieties are reported in the paper. In general, the results are interesting, especially the striking difference between the Pd/Py-COF and Pd/BE-COF materials?

This work might be of significance to the field of catalysis and related field of COF materials. For the

most part, it is comparable to the established literature in terms of scientific quality and novelty. However the experimental results on catalyst characterization do not provide convincing support for the conclusions and claims, especially with respect to the explanation of the difference between the Pd/Py-COF and Pd/BE-COF materials. The reasoning about non-covalent interactions are rather speculations than a sound explanation. Indeed, the data on IR spectra of adsorbed CO just show a different dispersion of Pd in the samples, while the band position of carbonyls is the same for both samples. Also, the XPS data demonstrate the same electronic state of Pd in these two samples with exactly the same binding energy of Pd. The difference of 0.1 eV is too low and can be explained by partial sample charging. Usually, larger shifts (0.3-0.5 eV) are considered in the literature as an indication of the changes in the electronic state of the supported metal. Obviously additional evidence is needed. These flaws prohibit publication and require a considerable revision of the manuscript. The methodology is sound for the most part, but the data on Pd dispersion (from CO or H₂ adsorption) should be added. It should be noted that enough details are provided in the methods for the work to be reproduced.

Response to the Reviewers' comments

Responses to Reviewer #1

Comments

In this manuscript, Yang, Luo and coworkers described Pd/Py-COF showed much higher activity than Pd/Be-COF and Pd/TB-COF in hydrogenation of AP to phenyl ethanol for the π - π interaction among pyrene and AP. Pyrene rings in the skeleton of COFs can greatly improve the activity of Pd NPs in the selective hydrogenation of a series of polar groups (C=O, NO₂, C=N) of aromatic compounds. Moreover, the characterizations of Pd NPs and a plausible mechanism were proposed. Hence, I would like to recommend this manuscript published in Nature Communications after the authors addressed the following issues:

Response: Thank you so much for your careful review and valuable suggestions. The questions are answered point-by-point as follows:

Question 1: "As the authors mentioned that the effect of pyrene group in hydrogenation of cyclohexanone to cyclohexanol was found by calculation, so it indicated the important role of π - π interactions play in this system. Except the calculation, cyclohexanone or one more example should be tested under laboratory conditions to further confirm the effect."

Response: Thank you for the suggestion. In addition to cyclohexanone, n-butylaldehyde was used as the substrate to confirm the role of π - π interaction. As shown in Supplementary Fig. 11, the almost coincided kinetic curves of Pd/Py-COF and Pd/Be-COF for n-butylaldehyde hydrogenation suggest the important role of π - π interaction in Pd/Py-COF catalyzed AP hydrogenation. The above discussions were added in the revised manuscript.

Supplementary Fig 11. Reaction profiles of supported Pd NPs in **a** CHO hydrogenation (130 °C, 20 bar of H₂, 0.07 mmol of CHO, Pd catalysts 4 mmol%, 2 mL of H₂O) and **b** n-butylaldehyde hydrogenation (80 °C, 20 bar of H₂, 0.07 mmol of butylaldehyde, Pd catalysts 4 mmol%, 2 mL H₂O).

Question 2: “Compare with entry 1 and 2 in Table 1: the strong interaction existed between pyrene rings and naphthalene ring, but it took longer time to access the final product? The similar phenomenon was found in entry 2 and 3. How about using the same reaction time to carry out these reactions?”

Response: Under identical reaction time (1 h), the conversion of the substrates on Pd/Py-COF follows the order of 2-naphthaldehyde > AP > 2-acetonaphthone. Similar trend for Pd/Be-COF was observed (Table 1).

Table 1. The conversion of the substrates on Pd/Py-COF with identical reaction time.

Substrate	Product	Conversion (%)	Selectivity (%)
		56	> 99
		20.3	> 99
		> 99	> 99

Reaction conditions: 40 °C, 10 bar of H₂, 2 mL of EtOH, 0.3 mmol of substrates, Pd/Py-COF 0.833mol%, 1h.

Generally, the activity of ketones on metal surface is lower than aldehydes due to the steric effect (Guo, M., et al. *ACS Catal.* **8**, 6476-6485 (2018)). Steric effect originates from the fact that each atom in a molecule occupies a certain amount of space. When atoms are brought together, hindrance will be induced in the expense of shape, energy, reactivity, etc (Liu, S., et al. *J. Chem. Phys.* **126**, 244103 (2007)). For example, the bulky groups slow down the reaction by limiting the accessibility of the reactive centre (Pinter, B., et al. *Phys. Chem. Chem. Phys.*, 2012, **14**, 9846-9854). Therefore, 2-naphthaldehyde is the most active molecule among the substrates. Compared with AP hydrogenation, the low activity of 2-acetonaphthone hydrogenation may arise from the balance between low intrinsic activity due to its large size molecule and the stronger π - π interaction between pyrene rings and naphthalene ring.

Question 3: “For entry 7, the hydrogenation of was similar to be more difficult by using the Pd catalysts the authors made, but the author did large scale to evaluate the efficiency of Pd/Be-COF. Providing the results by decreasing the amount of substrate to 0.3 mmol? the reaction condition of entry was missing?”

Response: Thank you for your nice reminding. For 2-acetonaphthone, 2-naphth aldehyde, benzofuran, quinoline and furfural, the used dosage of catalysts and substrates are identical to AP hydrogenation (40 °C, 10 bar of H₂, 2 mL of EtOH, 0.3 mmol of substrates, Pd catalysts 0.833mol%). For the substrates with high activity, the substrate amount was raised to 8 mmol and S/C ratio was also increased for N-benzylidenemethylamine (S/C = 2000), nitrobenzene (S/C = 5000) and styrene hydrogenation (S/C = 5000), respectively. We have added the reaction conditions in

the revised manuscript and supporting information.

Question 4: “Many hydrogenation catalysts were known, I suggest to conduct a positive control to further confirm the advantages of Pd/Be-COF analogues.”

Response: Thanks for your good suggest. The commercial Pd/C was used as model catalyst to elucidate the positive role of π - π interaction in activity enhancing with 1-aminopyrene (Py-NH₂) and aniline (AN) as the modifier. 1-Aminopyrene (Py-NH₂) and aniline (AN) were respectively added in AP hydrogenation using commercial Pd/C as model catalyst to understand the influence of the π - π interaction on the catalytic activity (Fig. 3e). This is one of the most frequently used method to tune the microenvironment of metal NPs^{7,10}. With 0.02 equiv. of Py-NH₂ with respect to the surface Pd, the conversion of AP increases sharply from 18.1% to 29.1%. The conversion reaches a maximum of 34.5% with 0.13 equiv. of Py-NH₂. The typical volcanic curve shows the positive effect of pyrene ring in enhancing the AP activity of Pd. The decrease in AP conversion at higher amounts of Py-NH₂ is due to the active sites capping. In contrast, the AP conversion decreases obviously in the presence of AN. The above results suggest that the π - π interaction could boost the AP activity of Pd/C, which further confirms that π - π interaction enhances the activity of Pd/Py-COF in AP hydrogenation. A slightly increased selectivity to phenyl ethanol (~97% vs. 99%) for Pd/C after Py-NH₂ and AN modification similar to previous reports that the amine ligands favoured high selectivity to the alcohol in carbonyl hydrogenation.^{38,40} We have added the above discussion in the revised manuscript.

Conversion on Pd/C with different amounts of 1-aminopyrene (Py-NH₂) and aniline (AN) in AP hydrogenation (Reaction conditions: 2.6×10^{-3} mmol of Pd, 2 mL of EtOH, 40 °C, 10 bar of H₂).

Responses to Reviewer #2

Comments

This paper demonstrates the importance of weak π - π interactions between the reactant and pyrene-containing COF (Py-COF: This also acts as a support of Pd nanoparticles) that would accelerate the catalytic selective hydrogenation

performances on Pd nanoparticles. The authors insist this promotion effect of π - π interactions by presenting several control experiments using Be-COF and TB-COF that can possess Pd nanoparticles of similar size and surface structure. The catalytic reaction results are interesting; however, I have several important questions/comments that need to be addressed, which is related to the authors insist. Therefore, I cannot recommend acceptance in its current form in Nature Communications.

Response: Thank you so much for your careful review and valuable suggestions. The questions are answered point-by-point as follows:

Question 1: “line 204-208. The authors assume that one possible reason for the stabilization of TS of Pd/Py-COF on O hydrogenation step is the π - π interaction between pyrene ring and acetophenone (AP). However, it is very difficult to figure out the modelled structure of Pd/Py-COF and Pd/Py-COF to see the π - π interaction. The authors should present all calculated structures so as the reader can understand this interaction.”

Response: Thank you for your suggestion. We have added all the calculated structures in Supplementary Fig. 7 in the revised manuscript. It is very hard to accurately calculate the π - π interaction promoted AP hydrogenation on Pd surface due to the complex reaction system. To simplify the calculation, a Pd₄ was loaded on Py-COF and Be-COF, respectively. Because of father distance of aromatic groups from the reactant in our Pd₄ models, the pyrene/benzene ring cannot directly influence the reaction process. However, DFT calculation results show that the energy barrier of O addition step of AP hydrogenation could be significant decreased via the interaction between phenyl ring and Pd atom. On the basis of our control experiments and the strong interaction between the benzene ring of AP molecule and pyrene rings around Pd surface, we can conclude that the Pd NPs confined in the Py-COF could accelerate the AP hydrogenation activity via π - π interaction.

Calculated energy profile of the O hydrogenation step of AP hydrogenation on

Pd₄/Be-COF and Pd₄/Py-COF.

Question 2: “line 213-225. The evidence of π - π interaction between the reactant and Py moiety of Py-COF seems to be weak; only FT-IR wavenumber shift of AP is discussed without appropriate literature citation and/or support of theoretical calculations. The authors calculated interaction energies between AP and Py-COF; however, there was also no appropriate literature citation.”

Response: Thanks for the nice reminding. The refs 44, 45 (Yang, D. Q., et al. *J. Phys. Chem. B* 109, 4481-4484 (2005). Zhang, Y., et al. *J. Nanosci. Nanotechnol.* 7, 2366-2375 (2007).) and ref 30 (Georgakilas, V., et al. *Chem. Rev.* 116, 5464-5519 (2016).) have been added in the revised manuscript.

Question 3: “line 234-247. The comparison of catalytic reaction data using cyclohexanone (CHO) without aromatic ring is very important to experimentally support the effect of π - π interaction. TOFs of CHO on Pd/Py-COF and Pd/Be-COF were similar; however, the experimental conditions are the same to those on AP? The experimental conditions and TOF calculation method should be explicitly presented. How about conversion/selectivity plot similar to Figure 3b? The color bars of TOF value for CHO hydrogenation and interaction energy on Figure 4d are very similar, and the authors should revise it. Also how about other ketones with aliphatic hydrocarbon chains?”

Response: The experiment conditions for CHO hydrogenation is not the same for AP hydrogenation due to low activity of CHO hydrogenation on Pd/COFs in ethanol. This may be due to the poison effect of EtOH to Pd for CHO hydrogenation (Li, X., et al. *Res. Chem. Intermediat.* 45, 1249-1262 (2019)). According to the previous report (He, J., et al. *J. Catal.* 309, 362-375 (2014)), H₂O could boost the C=O hydrogenation rate through the solvation of sorbed hydrogen (Cheng, G., et al. *Nat. Catal.* 4, 976-985 (2021)), therefore, the kinetic curves of CHO hydrogenation on Pd/COFs were carried out with H₂O as solvent. As shown in Supplementary Fig 11a, the initial TOFs are 27 and 26 h⁻¹ respectively for Pd/Py-COF and Pd/Be-COF, suggesting the similar activities on the two catalysts. Almost coincided kinetic curves of CHO hydrogenation also suggest Pd/COFs have similar catalytic ability to CHO hydrogenation. Similar trends were also obtained using n-butyraldehyde as the substrates although the activity of n-butyraldehyde hydrogenation is much higher than that of CHO. Furthermore, Pd/Py-COF shows higher activity than Pd/Be-COF (87% versus 22%) in the AP hydrogenation with H₂O as solvent (Supplementary Table 5), showing that H₂O does not alter the AP activity order of Pd/COFs. On the basis of the above results, we can conclude that the promoted activity of AP hydrogenation results from the non-covalent π - π interactions.

Supplementary Fig 11a. Reaction profiles of supported Pd NPs in **a** CHO hydrogenation (130 °C, 20 bar of H₂, 0.07 mmol of CHO, Pd catalysts 4 mmol%, 2 mL of H₂O) and **b** butyraldehyde hydrogenation (80 °C, 20 bar of H₂, 0.07 mmol of butyraldehyde, Pd catalysts 4 mmol%, 2 mL H₂O).

As you suggested, the quality of Fig. 4d was improved and has been divided into Fig. 4e and 4f for clarification.

The experimental conditions and kinetic curves were added in Supplementary Fig 11 and TOF calculation method was added in the Supplementary Method in the revised manuscript.

Question 4: “line 191-194, Figure 4b. It is difficult to figure out the calculation routes (O hydrogenation followed by C, or opposite sequence). The authors must improve the Figure 4b by adding several key structural images.”

Response: Thanks for your nice reminding. We have added the structural images in the Fig. 4b and the quality of all the Figures in the revised manuscript has been improved.

Question 5: “Characterization of Py-COF: The authors must present detailed characterization data if Py-COF is the new material. For instances, elemental analyses data in the experimental section.”

Response: Thanks for your good suggest. The C, H, N elemental analyses and TGA of Py-COF were conducted and the results have added in the revised manuscript.

Elemental analysis of Py-COF (%): C 79.6, H 5, N 9.6.

TGA results of Py-COF.

Question 6: “Overall: Figures (both in the main text and SI) are basically difficult to figure out. The authors have to make a great effort to improve the Figures so as to make readers easy to understand. For instances, peak attribution of FT-IR and peak legends XPS peak fitting are missing in Figure 2. Also, appropriate figure captions are required in some figures.”

Response: Thanks for your nice reminding. We have thoroughly improved the quality of the Figures both in the main text and SI. The missing peak attribution, legends and captions are also added in the Figures in the revised manuscript.

Question 7: “HRTEM of only Pd/Py-COF is presented in Figure S4, which is easy to understand Pd nanoparticles. It is difficult to figure out the Pd nanoparticles of 1.7 nm when Pd loading is 10 wt% from Figure S4b. The authors should present HRTEM images for other samples.”

Response: Thanks for your good suggest. The HRTEM images and HAADF-STEM images of Pd/Be-COF, Pd/TB-COF and Pd/Py-COF (10%) have been added in the revised manuscript.

a HRTEM image and **b** HAADF-STEM image of Pd/Be-COF.

a HRTEM image and **b** HAADF-STEM image of Pd/TB-COF. **c** HRTEM image and **d** HAADF-STEM image of Pd/Py-COF.

Question 8: “Experimental section: The equations to calculate conversion, selectivity, and TOF should be presented. The calibration method of the binding energy in XPS data must be presented. Experimental method for Table 1 should also be presented.”

Response: Thanks for reminding. The conversion, selectivity to each product and the apparent TOF were defined as follows:

$$\text{Conversion} = \frac{(\text{mmols of reactants})_{in} - (\text{mmols of reactants})_{out}}{(\text{mmols of reactants})_{in}} \times 100\%$$

$$\text{Selectivity} = \frac{\text{mmols of product } i}{\sum \text{mmols of products}} \times 100\%$$

$$\text{TOF}/h^{-1} = \frac{\text{amt of substrate}/\text{mmol}}{\text{amt of } \frac{\text{Pd}}{\text{mmol}} \times \text{Pd dispersion}} \times \frac{\text{Conversion } (\sim 10\%)}{t/h}$$

XPS results were calibrated by setting the C_{1s} adventitious carbon peak position to 284.6 eV. We have added the above discussion and experimental method for Table 1

in the revised manuscript and supporting information.

Question 9: “Notation that is not an abbreviation should be presented at the first appearance. For instances TOSS appears at line 83 without no full notation. The authors must check thoroughly the manuscript.”

Response: Thanks for your nice reminding. All the abbreviation in the revised manuscript has been notated at the first appearance.

Responses to Reviewer #3

Comments

The results of studying Pd/COF hydrogenation catalysts containing different organic moieties are reported in the paper. In general, the results are interesting, especially the striking difference between the Pd/Py-COF and Pd/Be-COF materials? This work might be of significance to the field of catalysis and related field of COF materials. For the most part, it is comparable to the established literature in terms of scientific quality and novelty.

Response: Thank you so much for your careful review and valuable suggestions. The questions are answered point-by-point as follows:

Question 1: However, the experimental results on catalyst characterization do not provide convincing support for the conclusions and claims, especially with respect to the explanation of the difference between the Pd/Py-COF and Pd/Be-COF materials. The reasoning about non-covalent interactions are rather speculations than a sound explanation.

Response: The FT-IR spectra of AP adsorbed on Pd/COFs were carefully analyzed. In comparison with the aromatic C-H vibration of AP at 1599 cm^{-1} and 1582 cm^{-1} , the merged aromatic C-H vibration peak for AP absorbed on Pd/Be-COF indicates that the activities of these vibrational modes are reduced/modified due to interaction of AP and Pd/Be-COF⁴⁴. A broad band of aromatic C-H vibration of AP at $\sim 1586\text{ cm}^{-1}$ could also be observed on Pd/Py-COF. In comparison with Pd/Be-COF, the red shift of aromatic C-H vibration of AP on Pd/Py-COF implies stronger interaction between AP and Pd/Py-COF⁴⁵. We added this discussion in the revised manuscript.

To further elucidate the important role of π - π interaction in activity enhancement, we did two control experiments. First, in addition to cyclohexanone, n-butylaldehyde was used as the substrate to confirm the role of π - π interaction. As shown in Supplementary Fig. 11, the almost coincided kinetic curves of Pd/Py-COFs and Pd/Be-COF for n-butylaldehyde hydrogenation suggest the important role of π - π interaction in Pd/Py-COF catalyzed AP hydrogenation.

Supplementary Fig 11a. Reaction profiles of supported Pd NPs in **a** CHO hydrogenation (130 °C, 20 bar of H₂, 0.07 mmol of CHO, Pd catalysts 4 mmol%, 2 mL of H₂O) and **b** butyraldehyde hydrogenation (80 °C, 20 bar of H₂, 0.07 mmol of butyraldehyde, Pd catalysts 4 mmol%, 2 mL H₂O).

Second, the commercial Pd/C was used as model catalyst to elucidate the positive role of π - π interaction in activity enhancing with 1-aminopyrene (Py-NH₂) and aniline (AN) as modifier. To further elucidate the activity difference of Pd/Py-COF and Pd/Be-COF, 1-aminopyrene (Py-NH₂) and aniline (AN) were respectively added in AP hydrogenation using commercial Pd/C as the model catalyst (Fig. 3e). This is one of the most frequently used method to tune the microenvironment of metal NPs^{7,10}. In the presence of 0.02 equiv. of Py-NH₂ with respect to the surface Pd atom, the conversion of AP increases sharply from 18.1% to 29.1%. The conversion reaches a maximum of 34.5% with 0.13 equiv. of Py-NH₂. The typical volcanic curve shows the positive effect of pyrene ring in enhancing the AP activity of Pd/C. The decrease in AP conversion at higher amounts of Py-NH₂ is due to the active sites capping. In contrast, the AP conversion decreases obviously in the presence of AN. The above results suggest that the π - π interaction could boost the AP activity of Pd/C, which further confirms that π - π interaction enhances the activity of Pd/Py-COF in AP hydrogenation. We have added the above discussion in the revised manuscript.

Conversion on Pd/C with different amounts of 1-aminopyrene (Py-NH₂) and aniline

(AN) in AP hydrogenation (Reaction conditions: 2.6×10^{-3} mmol of Pd, 2 mL of EtOH, 40 °C, 10 bar of H₂).

Question 2: “Indeed, the data on IR spectra of adsorbed CO just show a different dispersion of Pd in the samples, while the band position of carbonyls is the same for both samples.” “The methodology is sound for the most part, but the data on Pd dispersion (from CO or H₂ adsorption) should be added.”

Response: Thanks for your nice suggestion. The same band position of C=O on Pd/COFs is possibly related with similar surface electronic structure of Pd NPs supported on Py-COF and Be-COF, which is confirmed by XPS and CO stripping voltammetry results.

The different intensity of Pd/Py-COF and Pd/Be-COF may be due to the different transmittance on the two samples with different colour (brown and yellow respectively for Pd/Py-COF and Pd/Be-COF, see below). We added this in the revised manuscript. The CO and H₂ chemisorption show that Pd/Py-COF and Pd/Be-COF have similar Pd dispersion, which is confirmed by the HRTEM results.

The pictures of **a** tableted Pd/Py-COF and **b** tableted Pd/Be-COF.

The data on Pd dispersion (from CO or H₂ adsorption) were added in the revised manuscript in Supplementary Table 1 in the revised manuscript (see below).

Physical parameters and Pd 3d binding energies of Pd/COFs and commercial Pd/C.

Sample	BET surface area (m ² g ⁻¹)	Pd dispersion (%) ^a	Pd dispersion (%) ^b	Pd 3d _{5/2} (eV) ^c	Pd ⁰ /Pd ²⁺ (%) ^c	L/B ratio ^d
Pd/Py-COF	567	26.2	21.3	335.3	68/32	0.14
Pd/Be-COF	1139	26.1	20.4	335.3	68/32	0.15

^aData obtained from CO chemisorption results. ^bData obtained from H₂ chemisorption by a HOT method. ^cData obtained from XPS results. ^dCalculated based on in situ FT-IR of CO adsorption.

Question 3: “Also, the XPS data demonstrate the same electronic state of Pd in these two samples with exactly the same binding energy of Pd. The difference of 0.1 eV is too low and can be explained by partial sample charging. Usually, larger shifts (0.3-0.5 eV) are considered in the literature as an indication of the changes in the electronic state of the supported metal. Obviously additional evidence is needed.”

Response: In addition to XPS technique, we also detected the Pd surface electronic structure by CO stripping voltammetry technique. Similar results were obtained. In the literature, it was reported that the imine COFs could stabilize the small size Pd NPs via the coordination bonds (Ding, S. Y., et al. *J. Am. Chem. Soc.* 133. 19816-19822 (2011)). Generally, the coordination bonds are formed by donating electrons from the lone pair electrons of N to the Pd surface d orbitals. On the basis of the bonds character, the N ligands may donate instead of transfer electron to the surface Pd (Chen, G., et al. *Nat. Mater.* 15, 564-569 (2016).). Therefore, the N ligands modified metal NPs usually show smaller shift than the conventional oxides supported metal NPs. We added this in the revised manuscript.

REVIEWERS' COMMENTS

Reviewer #2 (Remarks to the Author):

The authors are appreciated for carefully addressing the review comments. The manuscript was largely improved. I have several comments/suggestions before publication.

(1, Regarding to question 5 at the first review) The authors present elemental analysis values of Py-COF in the experimental section. The authors should provide the calculated CHN values together to assure that the observed CHN values match the calculated values (The difference is usually within 0.3% for organic compounds.). The values should be presented in the first decimal place.

(2) The reaction conditions should be explicit in the figure captions of Fig. 3b and c. The authors should thoroughly check other parts in the manuscript and supporting information, too.

Reviewer #3 (Remarks to the Author):

The authors revised the manuscript and responded to the comments and issues raised. I am quite satisfied with the responses and the revision made. Now the paper can be accepted for publication.

Response to the Reviewers' comments

Responses to Reviewer #2

Comments

The authors are appreciated for carefully addressing the review comments. The manuscript was largely improved. I have several comments/suggestions before publication.

Response: Thank you so much for your careful review and valuable suggestions. The questions are answered point-by-point as follows:

(1, Regarding to question 5 at the first review) The authors present elemental analysis values of Py-COF in the experimental section. The authors should provide the calculated CHN values together to assure that the observed CHN values match the calculated values (The difference is usually within 0.3% for organic compounds.). The values should be presented in the first decimal place.

Response: Thanks for your nice reminding. The theoretical C, H and N content of Py-COF should be 81.6 % for C, 4.8% for H and 6.4%, respectively. We obtained the C, H, N content respectively of 79.6%, 5.0 % and 9.6 % with an O/N/H Analyzer (EMGA-930) and a C/S Analyzer (EMIA-8100). The difference is higher than 0.3% for C and H. This is due to the fact that Py-COF is a polymer and it is not possible to obtain the molecular structure as precise as the organic compounds. However, the experimental N content is much higher than the theoretical content (Entry 1, 9.6% vs. 6.4%). This may be due to the adsorbed N₂ in the micropores of Py-COF during the analysis process. The control sample, polystyrene, also gave 1.0% N content, confirming that the adsorbed N₂ affect the test results (Entry 3). To reduce the N₂ contamination, we conducted the N elemental experiment by using mixture of Py-COF and KBr instead of pure Py-COF. As shown in the Table below, the normalized N content of Py-COF is 6.5%, which is close to the theoretical value (Entry 2). We have already added the retested N element results and theoretically calculated values of C, H, N together with the analysis details in the revised manuscript and Supplementary Methods.

The H and N elemental analysis results.

Entry	Sample	H (%)	N (%)
1	Py-COF	5.0	9.6
2	Py-COF+KBr	5.0	6.5
3	polystyrene	Not detected	1.0

(2) The reaction conditions should be explicit in the figure captions of Fig. 3b and c. The authors should thoroughly check other parts in the manuscript and supporting information, too.

Response: Thanks for the nice reminding. The reaction conditions have been added in the Figure captions of Fig. 3b, 3c, 3e, 4a, 4c, 4f and Supplementary Fig 11, 12.

Responses to Reviewer #3

Comments

The authors revised the manuscript and responded to the comments and issues raised. I am quite satisfied with the responses and the revision made. Now the paper can be accepted for publication.

Response: Thank you so much for your careful review and valuable suggestions.